# Spelt Wheat: An Alternative for Sustainable Plant Production at Low N-Levels

**Eszter Sugár, Nándor Fodor \*, Renáta Sándor , Péter Bónis, Gyula Vida and Tamás Árendás**

Agricultural Institute, Centre for Agricultural Research, Brunszvik u. 2, 2462 Martonvásár, Hungary;
sugar.eszter@agrar.mta.hu (E.S.); sandor.renata@agrar.mta.hu (R.S.); bonis.peter@agrar.mta.hu (P.B.);
vida.gyula@agrar.mta.hu (G.V.); arendas.tamas@agrar.mta.hu (T.A.)
\* Correspondence: fodor.nandor@agrar.mta.hu; Tel.: +36-22-569-554

**Abstract:** Sustainable agriculture strives for maintaining or even increasing productivity, quality and economic viability while leaving a minimal foot print on the environment. To promote sustainability and biodiversity conservation, there is a growing interest in some old wheat species that can achieve better grain yields than the new varieties in marginal soil and/or management conditions. Generally, common wheat is intensively studied but there is still a lack of knowledge of the competitiveness of alternative species such as spelt wheat. The aim is to provide detailed analysis of vegetative, generative and spectral properties of spelt and common wheat grown under different nitrogen fertiliser levels. Our results complement the previous findings and highlight the fact that despite the lodging risk increasing together with the N fertiliser level, spelt wheat is a real alternative to common wheat for low N input production both for low quality and fertile soils. Vitality indices such as flag leaf chlorophyll content and normalized difference vegetation index were found to be good precursors of the final yield and the proposed estimation equations may improve the yield forecasting applications. The reliability of the predictions can be enhanced by including crop-specific parameters which are already available around flowering, beside soil and/or weather parameters.

**Keywords:** wheat; spelt; sustainable plant production; N-fertilisation; grain yield; vitality indices

## 1. Introduction

Climate-smart agriculture (CSA) strives for sustainable productivity, quality and economic viability while leaving a minimal foot print on the environment [1,2]. Despite the growing need for food and feed raw materials, crop yield is only one factor of the portfolio of the desired plant performances [3]. Plant genotypes developed on conventional tillage may not necessarily adapt to the changed cropping environment and new, specifically adapted genotypes may need to be developed [4]. To promote sustainability and biodiversity conservation, there is a growing interest in some old wheat species as well. Ruiz et al. [5] described some yield-related traits that have been identified as potential targets to achieve better grain yields of old wheat varieties in no-tillage and minimum tillage systems. Special attention is directed to the possible production of alternative cereals in organic production [6]. These species are nowadays rather produced for feed as alternatives to oats and barley. Ancient wheat genotypes that have the ability to maintain green leaf area ('stay green' traits) throughout grain filling are potential candidates for adapting and improving wheat for higher yield in arid and semi-arid regions. 'Stay green' is a vital characteristic associated with the capacity of the plant to maintain $CO_2$ assimilation and photosynthesis [7]. Because of the more frequent and more severe extreme weather conditions, the 'stay green' characteristic is especially important for breeders in producing more drought and/or heat tolerant crop species.

Spelt wheat (*Triticum aestivum* ssp. spelta L.), the oldest known wheat species cultivated in ancient Egypt and Italy, was as a result of spontaneous crossings of wild grasses. Reviving of spelt wheat production has started in the hilly and mountainous region of Central Europe and North America at the end of the 20th century [8]. It is an alternative crop, growing without any special soil related and climatic demands [9]. Spelt has the potential for low input production and adaptation to harsh ecological conditions and resistance to diseases [10]. Owing to its hulled grain and genetic polymorphism of its population, spelt is resistant to pests and diseases and hence suitable for organic production [6]. Spelt wheat and its products could serve as an abundant source of protein and a great proportion of soluble fibre emerging in the final spelt wheat products [11].

The identification of those factors which are determining the adaptation and nitrogen (N) utilization of spelt wheat is important for the successful introduction of the crop to a new environment in the comparison of non-fertilized and fertilized (100 kgN ha$^{-1}$) circumstances [12]. Several studies compared the productivity of spelt and common wheat in particular years. Most of them reported substantially higher yield of common wheat. The difference in yield often was as great as 60% in favour of common wheat [13] comparing low (6.8 kg ha$^{-1}$) and high (33.8 kg ha$^{-1}$) phosphorus supply. In the study of Jablonskyté-Raščé et al. [14] the average common wheat yield was 28% higher than that of the spelt wheat using ecological fertilizers. Budzynski et al. [15] reported 2.55 t ha$^{-1}$ higher yield potential average of common wheat than spelt in response to N rates. Some studies though reported that spelt was able to produce similar amount of yields as the common wheat (e.g., [16]). Probably because of the fact that climatic conditions of particular years, notably the climate × fertilisation interactions could significantly influence the grain yield of winter wheat [17]. However, there is still a lack of knowledge of the competitiveness of spelt grown at extensive or medium fertilisation levels.

Based on the results of Lazauskas et al. [18] we may assume that under low or moderate fertilisation inputs nitrogen will remain a major limiting factor for realizing high winter wheat yields in the coming decades. Nitrogen fertilisation directly or indirectly influences the LAI (leaf area index), degree of soil coverage by plants, leaf chlorophyll content, and other biophysical parameters, that can be characterized by vegetation indices, such as NDVI (normalized difference vegetation index) or SPAD (strongly correlated to chlorophyll content). Vegetation indices can be used as indicators of crop growth [19], nutrient status [20], and yield development [21]. Yield forecasting on the basis of vegetation indices acquired in the early stages of development can help farmers to make decisions about irrigation or additional fertilisation demand [22]. Normalized difference vegetation index have been widely used in agricultural remote sensing applications [22]. Leaf chlorophyll content (indexed e.g., by SPAD value) can be used as an accurate plant N status indicator. SPAD allows precise N fertilizer requirement calculations that are fundamental for enhancing N uptake efficiency [23,24]. A number of studies investigated the leaf growth of common wheat (e.g., [25,26]), but there are only a few data available regarding LAI changes of spelt wheat.

In addition for grain crops, harvest index (HI), the ratio of harvested grain to aboveground biomass, could be used as a measure of reproductive efficiency [27]. Although the effect of agronomical factors on HI of winter wheat was studied in a large number of works, there are just a few similar data for spelt wheat.

Because of the large inter-annual variability it is important to monitor the yield formation process of cereals in various years. More extensive data on yield formation of different wheat species may assist the spreading of production of alternative, even healthier cereals. The aim of this study is to provide a detailed analysis of vegetative, generative and spectral properties of spelt and common wheat grown under different N (from zero to moderate) levels.

## 2. Materials and Methods

The effect of nitrogen fertilisation on the yield and vitality parameters under various common and spelt winter wheat varieties was studied in parallel experiments in a split-plot design in four replications. The experiments were carried out in the years 2015/2016, 2017/2018 and 2018/2019 at

the Agricultural Institute of the Centre for Agricultural Research in Martonvásár (47°30′ N, 18°82′ E). The experiment was suspended for the 2016/2017 growing season, because of technical reasons. The N fertiliser doses (always applied in the form of ammonium-nitrate) were 0, 40, 80, 120 kg ha$^{-1}$ (designated as N0, N40, N80 and N120, respectively) in the main plots. The same dose (120 kg ha$^{-1}$) of phosphorus (P) and potassium (K) were given to every plot each year. Conventional tillage (no ploughing, only disk and cultivator use) was applied in the 0–20 cm soil layer after the PK fertilisation. By-products were always left on the field and incorporated in the soil. N fertiliser was applied in two splits: one-third before sowing (with PK) and two-third in early spring at tillering. Three genotypes of common wheat, Mv Kolo, Mv Marsall and Mv Kokárda, and spelt wheat, Mv Martongold, Franckenkorn and Mv Vitalgold, were sown in plots. All the genotypes except Franckenkorn (German origin) were breeded at Martonvásár. Around 9 m$^2$ (1.44 × 6 m) plots were used for each (N-level × variety) treatment. The chernozem soil of the experiment is non acidic loam with deep A horizon (Table 1).

**Table 1.** Main physical and chemical properties of the experimental plot at different layers at Martonvásár (Hungary) in 2018.

| Depth (cm) | 0–30 | 30–60 | 60–90 |
|---|---|---|---|
| Bulk density (g cm$^{-3}$) | 1.47 | 1.49 | 1.49 |
| Soil organic matter (%) | 2.82 | 2.02 | 1.39 |
| pH | 7.2 | 7.4 | 7.5 |
| Sand fraction (%) | 27 | 26 | 24 |
| Silt fraction (%) | 40 | 41 | 44 |
| Clay fraction (%) | 33 | 33 | 32 |

Owing to its favourable hydraulic properties (water holding capacity is 0.2 cm$^3$ cm$^{-3}$) and high soil organic matter content, based on the EU-SHG European Soil Database [28], the experiment site belongs to one of the most fertile regions of Central Europe.

Data of monthly precipitation and air temperature were recorded at the meteorological station at Martonvásár (Figure 1). The total amount of precipitation in the vegetative period (October–June) was ~30% lower in 2018/2019 (350 mm) than in the other two years (475 mm in 2015/2016 and 495 mm in 2017/2018) and ~16% lower than the 30 years' average (419 mm). The distribution of precipitation was less favourable for wheat owing to a prolonged dry period in March and April in 2015/2016 and 2018/2019, but the drought was compensated by high amount of precipitation (139 mm) in May 2019 (around flowering). The mean temperature during the vegetative period was similar during the three experimental years (8.6 °C in 2015/2016; 8.9 °C in 2017/2018 and 8.8 °C in 2018/2019) but considerably higher than the 30 years' average (7.3 °C). On the other hand, the course of the spring temperature was considerably different across the years especially in 2018 when the relatively cold February–March period (4 °C colder than the other two years) was followed by a relatively (3.5–4.5 °C) warmer April–May period.

Planting took place on 17 October 2015, 26 October 2017, and 17 October 2018 and the plots were harvested in the first decade of July in each year. Grain yield was estimated from the harvested plot yields and were converted to tons per hectare. Harvest index was estimated from plant samples of 0.5-m long sections taken before harvest.

LAI was measured by a non-destructive method using AccuPAR ceptometer [29] at flowering stage. Eight measurements were made below the canopy, four parallel and four across to the rows in each plot. The parallel and perpendicular measurements were averaged. The maximum LAI (LAI$_{max}$) values were measured in the third decade of May in each year.

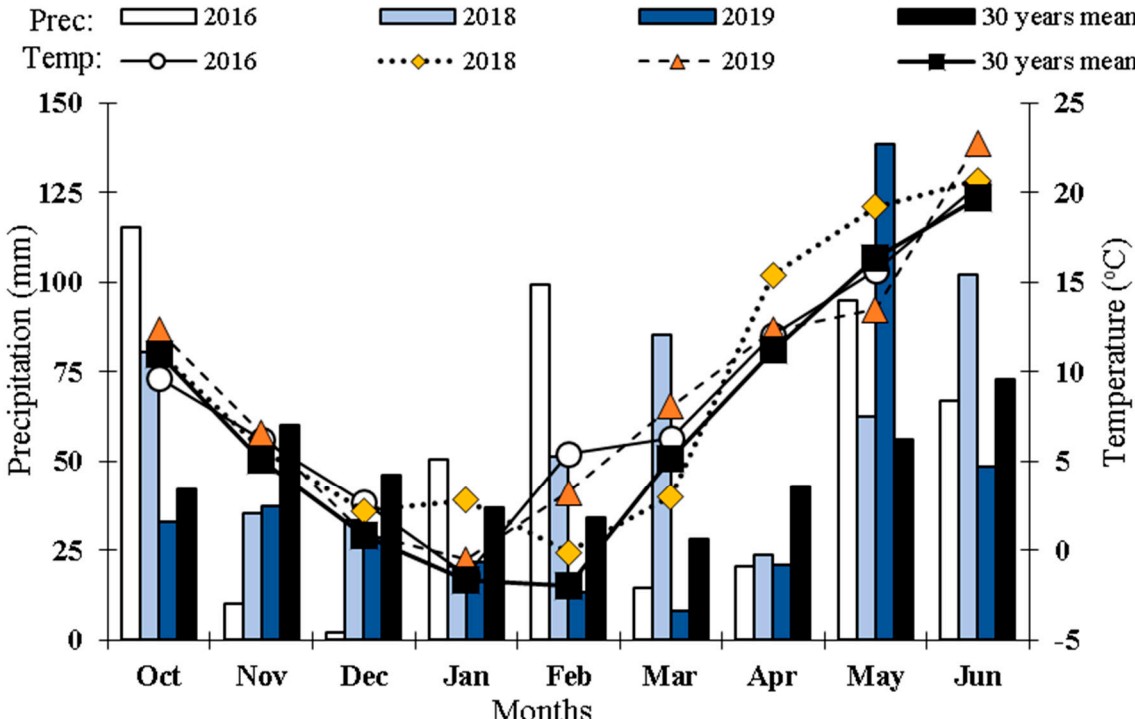

**Figure 1.** Monthly mean temperature and total precipitation at Martonvásár (Hungary) during the cropping seasons of 2015/2016, 2017/2018 and 2018/2019.

The chlorophyll content of the flag leaves at flowering was determined by using Minolta Chlorophyll Meter, SPAD-502 [30]. The measurements were made at the middle of the leaf lamina of 20 flag leaves. The SPAD values were converted to total chlorophyll values by using the conversion equation of Zhu et al. 2012 [31]. The 20 measurement results of each plot were averaged, and the mean values were used in the statistical analysis. NDVI was measured with a Trimble GreenSeeker® handheld crop sensor [32]. The measurements were made at flowering in sunny weather ~80 cm above the crop canopy. Two measurements per plot were carried out. The two readings in each plot were averaged, and the mean values were used in the statistical analysis.

The performance of spelt and common wheat, the effects of the different N fertiliser levels as well as the performance of the different varieties were evaluated with paired sample *t*-tests [33]. A difference was regarded to be significant in case the corresponding *t*-test resulted in a smaller than 0.05 probability (*p*) value.

Based on crop vitality indices (LAI$_{max}$, SPAD and NDVI) as independent variables, a multivariable linear yield estimation equation (model) was constructed (1) for both wheat species. This Equation (1) can be applied for yield (Y) forecast using data already available around flowering.

$$Y = a + b \times LAI_{max} + c \times SPAD + d \times NDVI \tag{1}$$

where a, b, c, d are fitting parameters, that were determined with regression analysis using the lm function of the stats v.3.6.1 R package [34].

From the 144 observed data record (Table A1 in the Appendix A) of the three years a random subset of 114 records were selected for determining/calibrating the parameters of the estimation equations (2). The remaining 30 records were used for validating the model. Estimated (Y$_e$) and observed (Y$_o$) yield data were compared using simple statistical indicators: Coefficient of determination (R$^2$) and mean absolute error (MAE), where the *mean* function calculates the arithmetic average of the arguments and *n* denotes the number of the estimated-observed data pairs.

$$R^2 = \frac{\left(\sum_{i=1}^{n}\left(Y_o^i - mean\left(Y_o^i\right)\right)\left(Y_e^i - mean\left(Y_e^i\right)\right)\right)^2}{\sum_{i=1}^{n}\left(Y_o^i - mean\left(Y_o^i\right)\right)^2 \sum_{i=1}^{n}\left(Y_e^i - mean\left(Y_e^i\right)\right)^2}$$

$$MAE = \sum_{i=1}^{n} \frac{|Y_o - Y_e|}{n}$$

(2)

## 3. Results and Discussion

All the observed data are presented in Table A1 in the Appendix A and summarized in Figures 2 and A1 in the Appendix A.

### 3.1. Grain Yield

When comparing the common and spelt wheat yields, statistically significant differences were found for each N-fertilisation levels, though the difference was only marginal in favour of common wheat at $N_{40}$ with a significance of $p = 0.031$. When pooling together the $N_0$ and $N_{40}$ yields for the three years spelt wheat had significantly higher production ($p = 0.033$) having 0.24 t ha$^{-1}$ higher average yield at this, low fertilisation level. At moderate fertilisation level ($N_{80}$ and $N_{120}$ together) common wheat had 1.14 t ha$^{-1}$ higher average yield that is a significant ($p \approx 0$) surplus compared to spelt. This result confirms that spelt wheat is a real alternative to common wheat for low input production [10] even for sites with fertile soils. Both common wheat and spelt had the highest yield under the maximal N dose in 2019, despite the fact that this was the driest experimental year. The high yield might be the result of the large amount of precipitation in May (~140 mm), that was ~82% higher than the multi-year average of that month. This underlines the importance of timing of the precipitation that might be an even more important factor in yield formation than the precipitation amount in certain years.

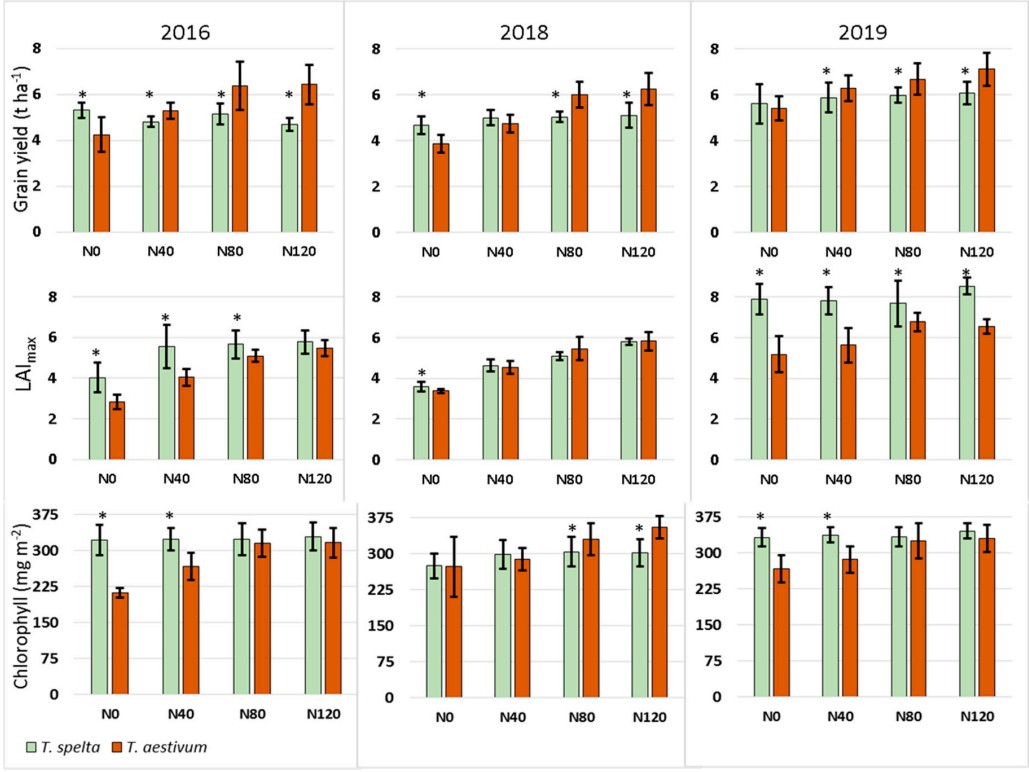

**Figure 2.** Harvested grain yield (t ha$^{-1}$), LAI$_{max}$ (m$^2$ m$^{-2}$) and chlorophyll content (mg m$^{-2}$) of spelt (*T. spelta*) and common wheat (*T. aestivum*) across varieties under four different nitrogen fertilisation treatments (0, 40, 80 and 120 kgN ha$^{-1}$) at Martonvásár (Hungary) in 2016, 2018 and 2019. * indicates statistically significant difference between spelt and common wheat.

The N-fertilisation significantly affected the grain yield (Figure 2) with a positive correlation between the N amount and the yield except for spelt in 2016. In 2016, high winds in June caused considerable lodging of growing degree with the increasing N fertilisation levels: 20, 45, 55 and 65% lodging at the $N_0$, $N_{40}$, $N_{80}$ and $N_{120}$ levels, respectively. Lodging made it very difficult for the harvester to properly harvest the plots resulting in uncertain and reduced yield results. In line with previous studies it is evident that lodging is clearly an issue in spelt production especially at higher N levels [35,36]. Common wheat showed much stronger reaction to the increasing fertiliser doses. Each increment in the N dose resulted in significantly higher yields. The $N_{120}$ common wheat yields were 31–61% higher than that of the $N_0$ yields. Even after excluding the 2016 data from the *t*-tests because of the lodging issue, spelt showed 8–9% yield increase when the $N_{120}$ yields were compared to the $N_0$ yields when yield averaged across 2018 and 2019. This is a moderate fertiliser effect, though statistically significant ($p = 0.0068$). There was a significant yield increase between the $N_0$ and $N_{40}$ levels ($p = 0.0092$) but the further N increments were not associated with further significant yield growth. According to this result spelt wheat can close in its yield potential even at very low fertiliser levels (approx. 40 kgN ha$^{-1}$ y$^{-1}$) on fertile soils. The variety selection had significant effect on the yield of both crops. Regarding the averaged N-treatments across the years, Mv Marsall, a common wheat variety had the highest yield (6.09 t ha$^{-1}$) that was significantly higher than the average yields of the other two common wheat varieties. Mv Martongold and Mv Vitalgold spelt wheat varieties provided the highest average yields (5.37 and 5.3 t ha$^{-1}$) that were significantly higher than the average yield of the Franckenkorn variety.

### 3.2. Harvest Index (HI)

The HI of the modern varieties of the intensively-cultivated grain crops is expected to fall within the range of 0.4 to 0.6 (40–60%) [37–39]. Considerably lower HI values were observed in our experiment for both crops: 33.1 to 44.0% for common wheat and 28.4 to 36.4% for spelt (Figure A1 in the Appendix A). In agreement with White and Wilson [38], N-fertilisation significantly increased the common wheat harvest index. The *t*-test resulted in a a $p = 0.03$ probability value when HI of the $N_0$ and $N_{40}$ levels were compared to the HI values of the $N_{80}$ and $N_{120}$ levels. In contrary, HI of spelt was the highest in the control treatment every year. The difference in HI was significant between the $N_0$ and $N_{40}$ fertilisation levels ($p = 0.039$) and even between the $N_{80}$ and $N_{120}$ levels ($p = 0.0001$). On average every 10 kg ha$^{-1}$ increase in the N fertiliser dose decreased the HI of common wheat with 0.3%. This result was in good agreement with previous findings that spelt is significantly more vigorous in tillering than standard bread wheat cultivars [13]. $LAI_{max}$ data (see Section 3.3) also confirms it.

### 3.3. $LAI_{max}$

$LAI_{max}$ values varied from 2.8 to 6.8 for common wheat and from 3.6 to 8.6 for spelt wheat (see Appendix A). Multi-year and multi-variety $LAI_{max}$ of spelt were 26.8, 22.8, 4.4 and 9.9% higher than that of common wheat across N fertilizer levels. These significant differences (corresponding *p* values were less than 0.027) clearly indicate the spelt is more vigorous in tillering, especially at low N levels. $LAI_{max}$ values grew significantly with the increasing N fertilisation level (Figure 2). Common wheat showed considerably more fertilisation-related $LAI_{max}$ growth. $LAI_{max}$ of the $N_{120}$ treatment was 52% higher than that of the $N_0$ treatment for common wheat while this difference was only 31% for spelt wheat. The observed LAI maximums of spelt wheat were considerably greater (even two times greater) than those reported in other studies [40], while the common wheat $LAI_{max}$ values were in good agreement with other studies [25,41]. Inter-annual variability could be a simple reason for this, as crop production could leave the so-called average range in certain years. Furthermore, results obtained at certain sites could be valid to other sites having different environmental conditions to a limited extent only. Thus, it is better to say that our results complement and do not contradict the previous findings on the maxima of spelt wheat leaf area index.

### 3.4. Chlorophyll Content of the Flag Leaf

The measured SPAD values (see Table A1 in the Appendix A) and the corresponding leaf chlorophyll contents overlapped with the equivalent values of forty winter wheat varieties investigated in an independent experiment at two nitrogen levels ($N_0$ and $N_{120}$) in three consecutive cropping seasons (2012/2013, 2013/2014 and 2014/2015) at Martonvásár, where the chlorophyll content ranged between 45 and 468 mg m$^{-2}$ [42]. In our experiment the chlorophyll content of spelt and common wheat ranged between 227 and 338 and 195 and 451 mg m$^{-2}$, respectively (Figure 2). Similarly to the yield, N fertilisation significantly increased the chlorophyll content of the flag leaf of both crops for every N dose increment with only one exception: the $N_{40} - N_{80}$ increment caused a non-significant increase in the spelt wheat chlorophyll content ($p = 0.48$). The chlorophyll content of common wheat showed a considerably stronger reaction to the increasing doses of N fertilisation. Spelt wheat chlorophyll contents were significantly higher than the common wheat chlorophyll contents in all three year at the $N_0$ and $N_{40}$ fertilisation levels. This again emphasizes the fact that spelt wheat has the capacity to use the resources of the soil more vigorously in limited environmental conditions.

### 3.5. Normalized Difference Vegetation Index (NDVI)

NDVI values (Table A1 and Figure A1 in the Appendix A), varied from 0.54 to 0.80 for common wheat and from 0.51 to 0.82 for spelt wheat, which were in good agreement with the measurements of Piekarczyk and Sulewska [22] who observed 0.72 and 0.71 average NDVI values for spelt and common wheat, respectively around flowering. Though there is a constant demand for deriving LAI data from NDVI, which is a standard component of remotely sensed datasets, the reality is that LAIs around and above 3 m$^2$ m$^{-2}$ are not distinguishable with NDVI data [43]. To make the issue even more complicated, according to our results, considerable interannual variable can be observed in the NDVI-LAI$_{max}$ correlation (Figure 3). The interannual difference is much more pronounced for spelt wheat, but the years 2016 and 2019 were considerably different for common wheat, as well. In general, there is certainly a positive correlation between the LAI$_{max}$ and the NDVI measured at flowering, but in certain years considerable deviations could be observed. If the 2016 and 2018 spelt wheat data are compared, similar LAI$_{max}$ values (5.2 and 4.8 m$^2$ m$^{-2}$; non-significant difference $p = 0.35$) correspond to significantly different NDVI values (0.54 and 0,72; $p \approx 0$). The difference between the two wheat species requires further investigation and highlights the fact that the NDVI-LAI interrelation is highly dependent on the plant species and probably on other environmental conditions as well.

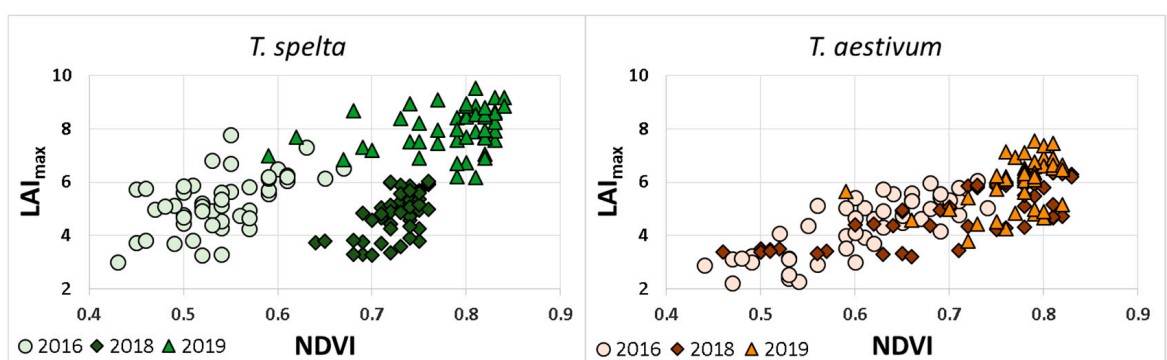

**Figure 3.** Correlation of leaf area index maximum (LAI$_{max}$) and NDVI values for spelt (*T.spelta*, left) and common wheat (*T. aestivum*, right) at Martonvásár (Hungary) in 2016 (dots), 2018 (diamonds) and 2019 (triangles).



*3.6. Multivariable Linear Yield Estimation Equation*

The linear regression (calibration) resulted in the following estimation equations for the two crops:

- Common wheat: $Y = 1.54155 + 0.63251 \times LAI_{max} + 0.02549 \times SPAD - 0.19102 \times NDVI$ ($R^2 = 0.5351$).
- Spelt wheat: $Y = 0.01856 + 0.09132 \times LAI_{max} + 0.07282 \times SPAD + 2.01997 \times NDVI$ ($R^2 = 0.4399$).

The equations were capable of estimating the yield with 0.64 and 0.37 t ha$^{-1}$ mean absolute error, that correspond to 11.2% and 7.1% relative errors for common and spelt wheat, respectively. When the equations were applied to the validation datasets the results were more moderate:

- Common wheat: $R^2 = 0.4557$; MAE = 0.81 t ha$^{-1}$.
- Spelt wheat: $R^2 = 0.4099$; MAE = 0.52 t ha$^{-1}$.

These kind of equations could be useful extensions to yield forecasting applications such as AgrometShell [44], since they add crop-specific parameters to the estimation beyond the already incorporated soil and weather-specific parameters.

## 4. Conclusions

A detailed analysis of vegetative, generative and spectral properties of spelt and common wheat grown under zero and moderate N levels was carried out at Martonvásár, Hungary in three cropping seasons. Our results extend the findings of Caballero et al. [10] and highlight the fact that despite the lodging risk increasing together with N fertiliser level, spelt wheat is a real alternative to common wheat for low N input production both for low quality and fertile soils. Spelt may help promoting sustainable crop production at sites where low input management is carried out because of any reasons by producing more yield than common wheat. Vitality indices such as flag leaf chlorophyll content and NDVI showed significant and moderate reaction to the increasing N fertiliser doses for common and spelt wheat. It was demonstrated that spelt wheat has considerably more moderate requirements compared to common wheat regarding soil nitrogen supply. Vitality indices were found to be good precursors of the final yield for both crops and the proposed estimation equations may improve the yield forecasting applications that use soil and/or weather parameters only by including crop-specific parameters that are already available around flowering.

**Author Contributions:** Conceptualization, T.Á., P.B. and E.S.; methodology, E.S. and G.V.; software, R.S.; validation, N.F., R.S. and E.S.; formal analysis, R.S. and N.F.; investigation, E.S. and P.B.; resources, G.V., P.B. and T.Á.; data curation, P.B.; writing—original draft preparation, E.S., R.S. and N.F.; writing—review and editing, E.S., R.S. and N.F.; visualization, R.S.; supervision, T.Á.; project administration, N.F.; funding acquisition, N.F.

**Funding:** This research was funded by the Széchenyi 2020 programme, and the European Regional Development Fund and the Hungarian Government, grant number GINOP-2.3.3-15-2016-00028.

**Conflicts of Interest:** The authors declare no conflict of interest.

## Appendix A

**Table A1.** NDVI, SPAD value, LAI$_{max}$ (m$^2$ m$^{-2}$), Harvest Index and harvested grain yield (t ha$^{-1}$), of spelt (*T. spelta*) and common wheat (*T. aestivum*) under four different nitrogen fertilization treatments (0, 40, 80 and 120 kgN ha$^{-1}$) at Martonvásár (Hungary) in 2016, 2018 and 2019.

| Year | N Level | Variety | Repetition | NDVI | | SPAD Value | | LAI$_{max}$ (m$^2$ m$^{-2}$) | | Harvest Index | | Yield (t/ha) | |
|---|---|---|---|---|---|---|---|---|---|---|---|---|---|
| | | | | Spelt | Aestivum | Spelt | Aestivum | Spelt | Aestivum | Spelt | Aestivum | Spelt | Aestivum |
| 2016 | 0 | 1 | 1 | 0.5 | 0.53 | 47.1 | 34.3 | 4.45 | 2.39 | 0.3 | 0.36 | 5.01 | 4.09 |
| 2016 | 0 | 1 | 2 | 0.54 | 0.56 | 50.9 | 35.7 | 3.29 | 2.92 | 0.32 | 0.32 | 5.39 | 4.02 |
| 2016 | 0 | 1 | 3 | 0.52 | 0.44 | 49.1 | 35 | 3.28 | 2.88 | 0.33 | 0.34 | 5.34 | 3.97 |
| 2016 | 0 | 1 | 4 | 0.51 | 0.53 | 51.2 | 34.8 | 3.82 | 3.15 | 0.3 | 0.33 | 5.59 | 3.95 |
| *2016* | *0* | *2* | *1* | *0.59* | *0.49* | *44.4* | *35.7* | *5.57* | *3.24* | *0.31* | *0.34* | *4.81* | *5.57* |
| *2016* | *0* | *2* | *2* | *0.57* | *0.6* | *46.8* | *36.9* | *4.27* | *3.01* | *0.29* | *0.28* | *5.18* | *5.71* |
| *2016* | *0* | *2* | *3* | *0.57* | *0.47* | *43.9* | *35.4* | *4.94* | *2.21* | *0.31* | *0.31* | *5.08* | *4.69* |
| *2016* | *0* | *2* | *4* | *0.54* | *0.47* | *43.1* | *35.7* | *4.3* | *3.12* | *0.3* | *0.3* | *4.91* | *4.51* |
| 2016 | 0 | 3 | 1 | 0.45 | 0.53 | 44.1 | 35.1 | 3.74 | 3.12 | 0.35 | 0.38 | 5.28 | 3.72 |
| 2016 | 0 | 3 | 2 | 0.43 | 0.54 | 45.3 | 33 | 3.01 | 2.27 | 0.33 | 0.4 | 5.53 | 3.57 |
| 2016 | 0 | 3 | 3 | 0.49 | 0.49 | 43.8 | 34 | 3.71 | 3.01 | 0.34 | 0.39 | 5.75 | 3.72 |
| 2016 | 0 | 3 | 4 | 0.46 | 0.53 | 43.1 | 33.3 | 3.83 | 2.54 | 0.33 | 0.38 | 5.86 | 3.39 |
| 2016 | 40 | 1 | 1 | 0.56 | 0.69 | 48.8 | 39 | 4.75 | 4.16 | 0.35 | 0.38 | 4.66 | 5.61 |
| 2016 | 40 | 1 | 2 | 0.57 | 0.61 | 50.8 | 39.8 | 4.66 | 3.93 | 0.33 | 0.35 | 5.13 | 5.82 |
| 2016 | 40 | 1 | 3 | 0.54 | 0.48 | 48 | 38.3 | 4.56 | 3.14 | 0.34 | 0.36 | 4.79 | 5.47 |
| 2016 | 40 | 1 | 4 | 0.59 | 0.59 | 48.2 | 39.9 | 5.7 | 4 | 0.34 | 0.37 | 4.97 | 5.87 |
| *2016* | *40* | *2* | *1* | *0.6* | *0.65* | *45.2* | *42.1* | *6.48* | *4.49* | *0.24* | *0.37* | *5.11* | *5.08* |
| *2016* | *40* | *2* | *2* | *0.55* | *0.62* | *44.5* | *46.7* | *7.76* | *4.64* | *0.29* | *0.43* | *4.43* | *5.53* |
| *2016* | *40* | *2* | *3* | *0.55* | *0.55* | *45.3* | *43* | *5.65* | *4.37* | *0.26* | *0.41* | *4.73* | *5.03* |
| *2016* | *40* | *2* | *4* | *0.63* | *0.6* | *46.1* | *44.5* | *7.3* | *4.12* | *0.27* | *0.39* | *4.73* | *5.24* |
| 2016 | 40 | 3 | 1 | 0.5 | 0.63 | 44.8 | 37.4 | 4.79 | 4.31 | 0.29 | 0.29 | 4.75 | 4.92 |
| 2016 | 40 | 3 | 2 | 0.52 | 0.59 | 45.7 | 39.8 | 5.21 | 3.53 | 0.28 | 0.33 | 4.86 | 5.14 |
| 2016 | 40 | 3 | 3 | 0.47 | 0.52 | 44.4 | 39.1 | 4.98 | 4.07 | 0.29 | 0.33 | 5.09 | 4.85 |
| 2016 | 40 | 3 | 4 | 0.53 | 0.62 | 42.9 | 38.2 | 4.92 | 3.71 | 0.3 | 0.34 | 4.43 | 4.88 |
| 2016 | 80 | 1 | 1 | 0.51 | 0.61 | 49.2 | 43 | 5.89 | 5.05 | 0.29 | 0.36 | 4.67 | 6.88 |
| 2016 | 80 | 1 | 2 | 0.53 | 0.71 | 51.5 | 41.2 | 4.4 | 4.77 | 0.29 | 0.39 | 5.37 | 7.14 |
| 2016 | 80 | 1 | 3 | 0.5 | 0.7 | 50.1 | 42.3 | 4.68 | 5.15 | 0.3 | 0.38 | 4.85 | 6.6 |
| 2016 | 80 | 1 | 4 | 0.49 | 0.6 | 49.3 | 41.4 | 5.14 | 5.41 | 0.28 | 0.37 | 5.93 | 6.95 |
| *2016* | *80* | *2* | *1* | *0.67* | *0.59* | *42.9* | *47.7* | *6.53* | *5.05* | *0.29* | *0.46* | *5.02* | *7.45* |
| *2016* | *80* | *2* | *2* | *0.55* | *0.64* | *43.9* | *48.7* | *6.71* | *5.57* | *0.3* | *0.43* | *4.39* | *7.29* |
| *2016* | *80* | *2* | *3* | *0.61* | *0.63* | *41.6* | *48.2* | *6.06* | *4.92* | *0.3* | *0.44* | *5.21* | *6.82* |
| *2016* | *80* | *2* | *4* | *0.59* | *0.66* | *43.9* | *47.3* | *6.16* | *5.34* | *0.29* | *0.45* | *4.78* | *7.25* |
| 2016 | 80 | 3 | 1 | 0.45 | 0.6 | 46.1 | 45.7 | 5.74 | 4.66 | 0.31 | 0.39 | 4.97 | 5.17 |
| 2016 | 80 | 3 | 2 | 0.52 | 0.71 | 45.7 | 46.3 | 5.11 | 5.31 | 0.32 | 0.36 | 5.35 | 5.44 |
| 2016 | 80 | 3 | 3 | 0.5 | 0.67 | 44.7 | 46.6 | 5.63 | 4.63 | 0.31 | 0.38 | 5.9 | 4.63 |
| 2016 | 80 | 3 | 4 | 0.46 | 0.56 | 44.6 | 47.2 | 5.77 | 5.13 | 0.32 | 0.37 | 5.24 | 4.76 |

**Table A1.** *Cont.*

| Year | N Level | Variety | Repetition | NDVI | | SPAD Value | | LAI$_{max}$ (m$^2$ m$^{-2}$) | | Harvest Index | | Yield (t/ha) | |
|---|---|---|---|---|---|---|---|---|---|---|---|---|---|
| | | | | Spelt | Aestivum | Spelt | Aestivum | Spelt | Aestivum | Spelt | Aestivum | Spelt | Aestivum |
| 2016 | 120 | 1 | 1 | 0.57 | 0.68 | 50.8 | 46.2 | 5.84 | 5.98 | 0.28 | 0.39 | 4.99 | 7.27 |
| 2016 | 120 | 1 | 2 | 0.52 | 0.66 | 51.7 | 45 | 4.92 | 5.58 | 0.32 | 0.38 | 4.66 | 7.26 |
| 2016 | 120 | 1 | 3 | 0.54 | 0.73 | 48.2 | 45.6 | 5.62 | 6.06 | 0.31 | 0.38 | 4.43 | 7.43 |
| 2016 | 120 | 1 | 4 | 0.54 | 0.74 | 49.9 | 43.4 | 5.12 | 5.04 | 0.3 | 0.4 | 4.67 | 7.16 |
| *2016* | *120* | *2* | *1* | *0.65* | *0.63* | *44.6* | *49.1* | *6.14* | *5.74* | *0.28* | *0.4* | *4.94* | *6.34* |
| *2016* | *120* | *2* | *2* | *0.59* | *0.66* | *45.8* | *49.3* | *6.21* | *5.3* | *0.25* | *0.45* | *4.46* | *7.06* |
| *2016* | *120* | *2* | *3* | *0.61* | *0.65* | *44.5* | *48.7* | *6.25* | *4.74* | *0.27* | *0.42* | *4.81* | *6.67* |
| *2016* | *120* | *2* | *4* | *0.61* | *0.71* | *45.1* | *47.9* | *6.19* | *5.8* | *0.29* | *0.4* | *4.49* | *6.63* |
| 2016 | 120 | 3 | 1 | 0.53 | 0.68 | 45.4 | 39.7 | 6.81 | 5 | 0.29 | 0.35 | 4.22 | 5.49 |
| 2016 | 120 | 3 | 2 | 0.54 | 0.69 | 45.4 | 45 | 5.36 | 5.46 | 0.28 | 0.31 | 5.09 | 5.62 |
| 2016 | 120 | 3 | 3 | 0.48 | 0.7 | 43.8 | 43.4 | 5.1 | 5.32 | 0.3 | 0.36 | 4.96 | 5.16 |
| 2016 | 120 | 3 | 4 | 0.5 | 0.69 | 45.5 | 42.8 | 5.85 | 5.56 | 0.29 | 0.41 | 4.44 | 5.11 |
| 2018 | 0 | 1 | 1 | 0.72 | 0.71 | 44.3 | 39.7 | 3.36 | 3.43 | 0.41 | 0.43 | 5.09 | 4 |
| 2018 | 0 | 1 | 2 | 0.69 | 0.63 | 45.2 | 57.6 | 3.31 | 3.3 | 0.32 | 0.38 | 5.12 | 3.58 |
| 2018 | 0 | 1 | 3 | 0.7 | 0.65 | 43.6 | 36.6 | 3.28 | 3.32 | 0.38 | 0.42 | 4.97 | 3.3 |
| 2018 | 0 | 1 | 4 | 0.68 | 0.66 | 44.1 | 40.7 | 3.29 | 3.22 | 0.35 | 0.4 | 4.89 | 3.85 |
| *2018* | *0* | *2* | *1* | *0.75* | *0.46* | *40.3* | *34.6* | *3.78* | *3.38* | *0.36* | *0.37* | *4.36* | *4.4* |
| *2018* | *0* | *2* | *2* | *0.71* | *0.5* | *40.2* | *37.6* | *3.69* | *3.51* | *0.32* | *0.38* | *4.23* | *4.36* |
| *2018* | *0* | *2* | *3* | *0.73* | *0.51* | *36.6* | *37.8* | *3.6* | *3.46* | *0.38* | *0.47* | *4.09* | *4.15* |
| *2018* | *0* | *2* | *4* | *0.74* | *0.57* | *38.6* | *37.6* | *3.9* | *3.4* | *0.37* | *0.41* | *3.98* | *3.54* |
| 2018 | 0 | 3 | 1 | 0.69 | 0.52 | 38.8 | 41.5 | 3.77 | 3.49 | 0.37 | 0.37 | 4.97 | 4.42 |
| 2018 | 0 | 3 | 2 | 0.64 | 0.5 | 41.4 | 44.6 | 3.74 | 3.38 | 0.39 | 0.42 | 4.86 | 3.68 |
| 2018 | 0 | 3 | 3 | 0.65 | 0.51 | 41.1 | 44.2 | 3.8 | 3.42 | 0.34 | 0.38 | 4.73 | 3.49 |
| 2018 | 0 | 3 | 4 | 0.68 | 0.56 | 42.7 | 40.4 | 3.81 | 3.32 | 0.38 | 0.44 | 4.76 | 3.62 |
| 2018 | 40 | 1 | 1 | 0.73 | 0.78 | 49.2 | 41.7 | 4.9 | 4.31 | 0.27 | 0.38 | 5.44 | 4.57 |
| 2018 | 40 | 1 | 2 | 0.75 | 0.75 | 46 | 43.2 | 4.94 | 4.22 | 0.34 | 0.33 | 5.59 | 4.06 |
| 2018 | 40 | 1 | 3 | 0.76 | 0.76 | 48.2 | 39.8 | 4.99 | 4.28 | 0.35 | 0.44 | 5 | 4.02 |
| 2018 | 40 | 1 | 4 | 0.71 | 0.72 | 46.3 | 42 | 5.03 | 4.34 | 0.32 | 0.47 | 5.23 | 5.04 |
| *2018* | *40* | *2* | *1* | *0.72* | *0.68* | *40.8* | *42.4* | *4.44* | *4.36* | *0.35* | *0.4* | *4.75* | *5.45* |
| *2018* | *40* | *2* | *2* | *0.74* | *0.6* | *41.7* | *39.9* | *4.58* | *4.41* | *0.32* | *0.43* | *4.87* | *4.78* |
| *2018* | *40* | *2* | *3* | *0.71* | *0.62* | *44.2* | *39.3* | *4.66* | *4.44* | *0.37* | *0.45* | *4.47* | *4.97* |
| *2018* | *40* | *2* | *4* | *0.74* | *0.64* | *42.6* | *44.1* | *4.73* | *4.37* | *0.33* | *0.47* | *4.7* | *4.87* |
| 2018 | 40 | 3 | 1 | 0.75 | 0.69 | 42 | 44.4 | 4.26 | 4.94 | 0.32 | 0.38 | 5.06 | 4.9 |
| 2018 | 40 | 3 | 2 | 0.74 | 0.7 | 42.2 | 45.3 | 4.33 | 5.06 | 0.39 | 0.44 | 5.22 | 4.78 |
| 2018 | 40 | 3 | 3 | 0.7 | 0.65 | 41.5 | 46.2 | 4.58 | 4.89 | 0.35 | 0.46 | 4.9 | 4.81 |
| 2018 | 40 | 3 | 4 | 0.72 | 0.65 | 41 | 44.9 | 4.27 | 4.95 | 0.36 | 0.49 | 4.76 | 4.61 |

**Table A1.** *Cont.*

| Year | N Level | Variety | Repetition | NDVI | | SPAD Value | | $LAI_{max}$ $(m^2\ m^{-2})$ | | Harvest Index | | Yield (t/ha) | |
|------|---------|---------|------------|------|------|------|------|------|------|------|------|------|------|
| | | | | Spelt | Aestivum | Spelt | Aestivum | Spelt | Aestivum | Spelt | Aestivum | Spelt | Aestivum |
| 2018 | 80 | 1 | 1 | 0.73 | 0.81 | 48.6 | 43.9 | 5.27 | 4.66 | 0.33 | 0.4 | 5.19 | 5.45 |
| 2018 | 80 | 1 | 2 | 0.74 | 0.81 | 46.3 | 46.9 | 5.21 | 4.69 | 0.35 | 0.42 | 5.16 | 5.03 |
| 2018 | 80 | 1 | 3 | 0.74 | 0.82 | 47.9 | 42.9 | 5.32 | 4.71 | 0.35 | 0.43 | 5.01 | 5.49 |
| 2018 | 80 | 1 | 4 | 0.75 | 0.81 | 49.2 | 47 | 5.36 | 4.69 | 0.34 | 0.41 | 5.48 | 6.23 |
| *2018* | *80* | *2* | *1* | *0.72* | *0.72* | *43.1* | *46.3* | *5.12* | *5.85* | *0.33* | *0.35* | *5.13* | *6.27* |
| *2018* | *80* | *2* | *2* | *0.74* | *0.73* | *40.7* | *42.8* | *5.07* | *5.79* | *0.36* | *0.45* | *4.87* | *6.17* |
| *2018* | *80* | *2* | *3* | *0.75* | *0.75* | *40.8* | *44.7* | *5.16* | *5.83* | *0.34* | *0.45* | *4.75* | *5.86* |
| *2018* | *80* | *2* | *4* | *0.72* | *0.76* | *42* | *44.7* | *5.13* | *5.88* | *0.32* | *0.46* | *4.59* | *7.09* |
| 2018 | 80 | 3 | 1 | 0.69 | 0.75 | 44 | 51.1 | 4.85 | 5.93 | 0.33 | 0.42 | 5.09 | 6.13 |
| 2018 | 80 | 3 | 2 | 0.72 | 0.75 | 43.2 | 51.1 | 4.91 | 5.89 | 0.34 | 0.49 | 5.12 | 6.13 |
| 2018 | 80 | 3 | 3 | 0.71 | 0.79 | 42.7 | 48.5 | 4.81 | 5.9 | 0.35 | 0.44 | 4.97 | 6.63 |
| 2018 | 80 | 3 | 4 | 0.73 | 0.73 | 43.3 | 51.5 | 4.88 | 5.85 | 0.34 | 0.43 | 5.08 | 5.64 |
| 2018 | 120 | 1 | 1 | 0.74 | 0.82 | 45.3 | 48.1 | 5.8 | 6.31 | 0.32 | 0.37 | 5.39 | 5.21 |
| 2018 | 120 | 1 | 2 | 0.76 | 0.81 | 46.2 | 47.5 | 5.92 | 6.36 | 0.29 | 0.46 | 5.58 | 5.07 |
| 2018 | 120 | 1 | 3 | 0.74 | 0.83 | 48.6 | 46.3 | 5.86 | 6.29 | 0.3 | 0.39 | 5.33 | 6.38 |
| 2018 | 120 | 1 | 4 | 0.72 | 0.83 | 49.2 | 47.8 | 5.98 | 6.2 | 0.28 | 0.4 | 5.62 | 6.78 |
| *2018* | *120* | *2* | *1* | *0.73* | *0.76* | *41.8* | *48.7* | *5.88* | *6.01* | *0.27* | *0.45* | *3.86* | *5.97* |
| *2018* | *120* | *2* | *2* | *0.72* | *0.78* | *43.3* | *48.5* | *5.99* | *5.88* | *0.27* | *0.49* | *4.76* | *6.23* |
| *2018* | *120* | *2* | *3* | *0.76* | *0.8* | *42.9* | *46.9* | *5.93* | *5.79* | *0.29* | *0.47* | *4.22* | *6.16* |
| *2018* | *120* | *2* | *4* | *0.76* | *0.73* | *42.1* | *48.5* | *6.03* | *5.88* | *0.25* | *0.46* | *5.03* | *7.3* |
| 2018 | 120 | 3 | 1 | 0.73 | 0.78 | 42.3 | 50.6 | 5.57 | 5.11 | 0.37 | 0.43 | 5.21 | 5.97 |
| 2018 | 120 | 3 | 2 | 0.75 | 0.79 | 41 | 52.7 | 5.67 | 5.48 | 0.32 | 0.44 | 5.48 | 6.2 |
| 2018 | 120 | 3 | 3 | 0.75 | 0.81 | 43.6 | 52.6 | 5.59 | 5.16 | 0.34 | 0.46 | 5.33 | 7.33 |
| 2018 | 120 | 3 | 4 | 0.74 | 0.78 | 42.2 | 51.7 | 5.68 | 5.5 | 0.38 | 0.46 | 5.42 | 6.42 |
| 2019 | 0 | 1 | 1 | 0.59 | 0.76 | 49.4 | 40.5 | 6.98 | 6.16 | 0.29 | 0.35 | 4.28 | 5.01 |
| 2019 | 0 | 1 | 2 | 0.67 | 0.76 | 49.4 | 41.1 | 6.84 | 6.2 | 0.35 | 0.31 | 5.49 | 5.69 |
| 2019 | 0 | 1 | 3 | 0.83 | 0.76 | 49.9 | 39.8 | 7.56 | 6.13 | 0.32 | 0.3 | 5.92 | 4.76 |
| 2019 | 0 | 1 | 4 | 0.77 | 0.78 | 48.1 | 40.4 | 7.45 | 6.22 | 0.31 | 0.32 | 6.53 | 5.98 |
| *2019* | *0* | *2* | *1* | *0.68* | *0.66* | *45.5* | *39.1* | *8.66* | *4.59* | *0.31* | *0.31* | *5.07* | *5.6* |
| *2019* | *0* | *2* | *2* | *0.73* | *0.59* | *45.8* | *38* | *8.38* | *5.64* | *0.32* | *0.33* | *6.15* | *5.9* |
| *2019* | *0* | *2* | *3* | *0.83* | *0.72* | *45* | *37.7* | *9.18* | *5.41* | *0.33* | *0.34* | *5.96* | *6.14* |
| *2019* | *0* | *2* | *4* | *0.81* | *0.7* | *45.8* | *35.6* | *8.85* | *4.98* | *0.28* | *0.37* | *6.3* | *5.3* |
| 2019 | 0 | 3 | 1 | 0.62 | 0.73 | 45.7 | 44.1 | 7.7 | 4.43 | 0.32 | 0.39 | 4.63 | 4.37 |
| 2019 | 0 | 3 | 2 | 0.69 | 0.72 | 46.7 | 43.8 | 7.32 | 3.79 | 0.31 | 0.35 | 4.26 | 5.19 |
| 2019 | 0 | 3 | 3 | 0.82 | 0.76 | 45.2 | 44.5 | 7.7 | 4.26 | 0.35 | 0.36 | 6.03 | 5.3 |
| 2019 | 0 | 3 | 4 | 0.75 | 0.75 | 48.2 | 43.3 | 8.2 | 4.52 | 0.32 | 0.35 | 6.74 | 5.81 |

**Table A1.** *Cont.*

| Year | N Level | Variety | Repetition | NDVI | | SPAD Value | | LAI$_{max}$ (m$^2$ m$^{-2}$) | | Harvest Index | | Yield (t/ha) | |
|---|---|---|---|---|---|---|---|---|---|---|---|---|---|
| | | | | Spelt | Aestivum | Spelt | Aestivum | Spelt | Aestivum | Spelt | Aestivum | Spelt | Aestivum |
| 2019 | 40 | 1 | 1 | 0.74 | 0.81 | 50.9 | 42.1 | 7.52 | 6.75 | 0.35 | 0.36 | 5.61 | 5.81 |
| 2019 | 40 | 1 | 2 | 0.81 | 0.8 | 48.2 | 44.5 | 7.9 | 6.92 | 0.26 | 0.41 | 5.63 | 5.9 |
| 2019 | 40 | 1 | 3 | 0.82 | 0.79 | 48.4 | 44.5 | 7.66 | 6.57 | 0.27 | 0.36 | 6.58 | 5.45 |
| 2019 | 40 | 1 | 4 | 0.79 | 0.8 | 48.3 | 42.1 | 7.56 | 6.6 | 0.32 | 0.34 | 5.89 | 6.65 |
| *2019* | *40* | *2* | *1* | *0.79* | *0.77* | *45.4* | *39.5* | *8.41* | *4.83* | *0.35* | *0.37* | *6.07* | *6.53* |
| *2019* | *40* | *2* | *2* | *0.8* | *0.78* | *48.4* | *40.6* | *8.53* | *5.63* | *0.33* | *0.4* | *5.92* | *6.95* |
| *2019* | *40* | *2* | *3* | *0.83* | *0.75* | *46.1* | *40.1* | *8.64* | *5.74* | *0.25* | *0.37* | *6.38* | *6.91* |
| *2019* | *40* | *2* | *4* | *0.82* | *0.79* | *46.2* | *40.2* | *8.76* | *4.81* | *0.33* | *0.39* | *6.17* | *6.88* |
| 2019 | 40 | 3 | 1 | 0.7 | 0.79 | 46.9 | 42.7 | 7.2 | 4.99 | 0.32 | 0.33 | 4.16 | 5.32 |
| 2019 | 40 | 3 | 2 | 0.79 | 0.8 | 47.4 | 43.8 | 7.99 | 4.67 | 0.36 | 0.33 | 5.39 | 6.03 |
| 2019 | 40 | 3 | 3 | 0.82 | 0.82 | 46.3 | 42.2 | 7.04 | 5.17 | 0.35 | 0.35 | 6.17 | 6.31 |
| 2019 | 40 | 3 | 4 | 0.8 | 0.8 | 47.4 | 49.2 | 6.74 | 4.91 | 0.36 | 0.35 | 6.59 | 6.52 |
| 2019 | 80 | 1 | 1 | 0.75 | 0.81 | 46.6 | 46.4 | 6.91 | 6.9 | 0.27 | 0.32 | 5.68 | 5.34 |
| 2019 | 80 | 1 | 2 | 0.75 | 0.79 | 49.9 | 45.4 | 7.5 | 6.76 | 0.3 | 0.3 | 6.19 | 6.13 |
| 2019 | 80 | 1 | 3 | 0.81 | 0.82 | 48.3 | 45.4 | 6.17 | 6.52 | 0.33 | 0.3 | 6.02 | 6.59 |
| 2019 | 80 | 1 | 4 | 0.79 | 0.82 | 49.9 | 47.5 | 6.2 | 6.59 | 0.34 | 0.28 | 5.91 | 6.8 |
| *2019* | *80* | *2* | *1* | *0.81* | *0.78* | *45.6* | *41.9* | *9.51* | *6.06* | *0.31* | *0.35* | *6.18* | *6.32* |
| *2019* | *80* | *2* | *2* | *0.74* | *0.78* | *44.9* | *41.1* | *8.93* | *6.53* | *0.28* | *0.34* | *5.88* | *7.07* |
| *2019* | *80* | *2* | *3* | *0.83* | *0.79* | *44.5* | *43.6* | *7.93* | *6.36* | *0.31* | *0.35* | *6.06* | *6.92* |
| *2019* | *80* | *2* | *4* | *0.82* | *0.78* | *47.3* | *44* | *7.96* | *6.4* | *0.34* | *0.36* | *6.7* | *7.41* |
| 2019 | 80 | 3 | 1 | 0.8 | 0.79 | 49.3 | 47.3 | 8.43 | 7.53 | 0.3 | 0.36 | 5.72 | 6.76 |
| 2019 | 80 | 3 | 2 | 0.77 | 0.81 | 46.6 | 50 | 9.07 | 6.91 | 0.35 | 0.37 | 5.51 | 7.79 |
| 2019 | 80 | 3 | 3 | 0.82 | 0.8 | 46.1 | 51.7 | 6.9 | 7.37 | 0.33 | 0.37 | 6.3 | 5.81 |
| 2019 | 80 | 3 | 4 | 0.79 | 0.81 | 46.5 | 51.4 | 6.7 | 7.44 | 0.32 | 0.38 | 5.6 | 7.29 |
| 2019 | 120 | 1 | 1 | 0.77 | 0.81 | 50.7 | 45.6 | 7.94 | 6.53 | 0.27 | 0.34 | 6.07 | 5.98 |
| 2019 | 120 | 1 | 2 | 0.81 | 0.81 | 49.2 | 44.3 | 8.54 | 6.64 | 0.26 | 0.33 | 5.09 | 6 |
| 2019 | 120 | 1 | 3 | 0.83 | 0.82 | 49.9 | 46.1 | 8.23 | 6.64 | 0.3 | 0.3 | 5.49 | 6.95 |
| 2019 | 120 | 1 | 4 | 0.82 | 0.82 | 49.9 | 46.3 | 8.45 | 6.46 | 0.28 | 0.28 | 6.94 | 6.99 |
| *2019* | *120* | *2* | *1* | *0.83* | *0.79* | *47.2* | *45.1* | *8.58* | *6.35* | *0.33* | *0.33* | *6.28* | *6.41* |
| *2019* | *120* | *2* | *2* | *0.8* | *0.79* | *47.7* | *45.6* | *7.69* | *6.08* | *0.32* | *0.31* | *5.75* | *7.3* |
| *2019* | *120* | *2* | *3* | *0.84* | *0.78* | *46.1* | *43.4* | *9.17* | *6.31* | *0.29* | *0.31* | *6.04* | *7.85* |
| *2019* | *120* | *2* | *4* | *0.82* | *0.79* | *47.4* | *45* | *8.53* | *6.19* | *0.25* | *0.32* | *6.4* | *8.31* |
| 2019 | 120 | 3 | 1 | 0.8 | 0.75 | 48.6 | 50.7 | 8.85 | 6.24 | 0.29 | 0.36 | 6.17 | 7.5 |
| 2019 | 120 | 3 | 2 | 0.8 | 0.76 | 49 | 50.9 | 8.94 | 7.14 | 0.26 | 0.33 | 5.96 | 7.33 |
| 2019 | 120 | 3 | 3 | 0.84 | 0.77 | 46.7 | 49 | 8.86 | 6.93 | 0.28 | 0.39 | 6.59 | 7.12 |
| 2019 | 120 | 3 | 4 | 0.82 | 0.78 | 47 | 50.3 | 8.78 | 7.11 | 0.27 | 0.38 | 6.14 | 7.63 |

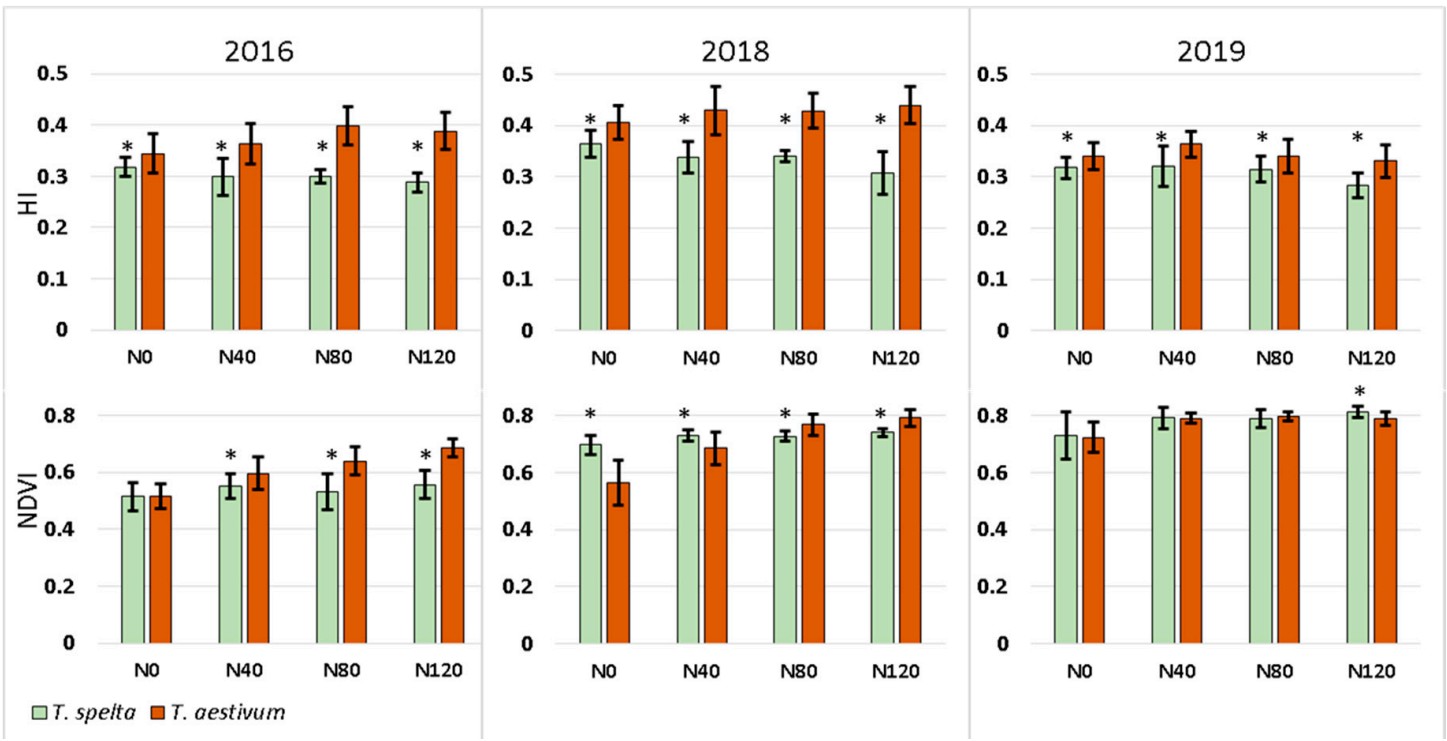

**Figure A1.** Harvest Index (HI) and NDVI of spelt (*T. spelta*) and common wheat (*T. aestivum*) across varieties under four different nitrogen fertilisation treatments (0, 40, 80 and 120 kgN ha$^{-1}$) at Martonvásár (Hungary) in 2016, 2018 and 2019. * indicates statistically significant difference between spelt and common wheat.

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
