# Peer review of "Spelt Wheat: An Alternative for Sustainable Plant Production at Low N-Levels"

_sustainability, doi:10.3390/su11236726_

Round 1

Reviewer 1 Report

major commments:

1) The title of the manuscript states that 'Spelt Wheat: an Alternative for Sustainable Plant Production at Moderate N-levels', but through the whole manuscript and at the conclusion, authors state that 'despite the lodging risk increasing together with N fertiliser level, spelt wheat is a real alternative to common wheat for low N input production both for low quality and fertile soils'. Moderate N levels or low N level? in the study, low N levels represent 0-40, while moderate N levels represent >40 if i am not mistaken. please clarify.

2) overall, the presentation of data (figures) is a bit overwhelming, authors should try to present the most related data. 

specific comments:

1) Line 31-34, P1: in abstract, too long sentence, try to break down

2) line 48, P2: add 'that' between the words 'genotypes' and 'have'

3) line 51, p2: change CO2 to CO2

4) line 53, p2: change 'droungth' to 'drought'

5) line 62, p2: change 'emerges' to 'emerging' 

6) line 70, p2: change 'than' to 'to'

7) line 65- 70, p2: do these studies cite here have different N levels? authors should mention that.

8) line 74-77, authors mention and present data on harvest index, but thourghout the manuscript, the data were not related to yield or used in the model except presenting the data. Try to remove any part related to harvest index in the manuscript or if authors want to keep it, move this paragraph after line 91, starting with words' in addition'.

9) line 87-89, p2: reword the sentence, it is too long.

10) line 94, p2: missing a transition sentence or word.

11) line 119, p3: i don't think 'ca.' is commonly used in paper, try to change it to 'approximately', 'about' etc. throughout the manuscript.

12) line 139, p4: change 'is' to 'in'

13) line 152-154, p4: break down the sentence. 

14) line 175-178, p5: looking at the figure 2, the yield from 2019 might not necessarily significantly higher under maximum N comparing to other two years. To make it clear, authors should add significant letters on the graph, it can be marked between spelt and common wheat, or between different N levels. 

15) Figure 2, are these values averaged across cultivars? if so, authors should state that in the manuscript.

16) line 190, p6: authors first introduce the common wheat as 'CW', should spell it out in parenthesis. change 'CW' to 'CW (Common Wheat)'. same as 'SW' when first time introducing it as spelt wheat.

17) line 192, p7: after excluding 2016 data, are the yield averaged between 2018 and 2019? authors should be more specific.

18) line 207, p7: change 'probabilty' to 'probability'

19) line 215-216, p7: 'LAImax of 215 SW was 26.8, 22.8, 4.4 and 9.9% higher than that of CW.' is this statement  for comparson across years and across cultivars? if so, authors should state that.

20) line 240, p8: change 'emphasize' to 'emphasizes' 

21) line 282-285, p9: too long sentence, try to break down.

Author Response

Please see our reply in the attached document.

Reviewer 2 Report

line 38: if you refer to Conservation Agriculture (CA) as defined by FAO (minimum soil disturbance/no-till + permanent soil cover + diversity of crops), pls. write the term with capital letters to show that you refer to a defined technical term.

line 99: while there is mentioning of Conservation Agriculture in the beginning, there is no further reference to it in the rest of the paper; in materials and methods there is not even an indication of the cropping system, in which the experiments are carried out (tillage or no-till, CA or conventional, crop rotation, residue cover, if CA, how long has the field been under CA?)

Author Response

Please find our reply in the attached document.

Round 2

Reviewer 2 Report

no further comments